# A multivariate statistical evaluation of actual use of electronic health record systems implementations in Kenya

**Philomena Ngugi** [1,2]*, **Ankica Babic**[1,3], **Martin C. Were**[4,5]

**1** Department of Information Science and Media studies, University of Bergen, Bergen, Norway, **2** Institute of Biomedical Informatics, Moi University, Eldoret, Kenya, **3** Department of Biomedical Engineering, Linköping University, Linköping, Sweden, **4** Department of Biomedical Informatics, Vanderbilt University Medical Center, Nashville, TN, United States of America, **5** Vanderbilt Institute of Global Health, Vanderbilt University Medical Center, Nashville, TN, United States of America

\* waruharip@gmail.com

## Abstract

### Background

Health facilities in developing countries are increasingly adopting Electronic Health Records systems (EHRs) to support healthcare processes. However, only limited studies are available that assess the actual use of the EHRs once adopted in these settings. We assessed the state of the 376 KenyaEMR system (national EHRs) implementations in healthcare facilities offering HIV services in Kenya.

### Methods

The study focused on seven EHRs use indicators. Six of the seven indicators were programmed and packaged into a query script for execution within each KenyaEMR system (KeEMRs) implementation to collect monthly server-log data for each indicator for the period 2012–2019. The indicators included: *Staff system use*, *observations (clinical data volume)*, *data exchange*, *standardized terminologies*, *patient identification*, and *automatic reports*. The seventh indicator (*EHR variable Completeness*) was derived from routine data quality report within the EHRs. Data were analysed using descriptive statistics, and multiple linear regression analysis was used to examine how individual facility characteristics affected the use of the system.

### Results

213 facilities spanning 19 counties participated in the study. The mean number of authorized users who actively used the KeEMRs was 18.1% (SD = 13.1%, p<0.001) across the facilities. On average, the volume of clinical data (*observations*) captured in the EHRs was 3363 (SD = 4259). Only a few facilities(14.1%) had health data exchange capability. 97.6% of EHRs concept dictionary terms mapped to standardized terminologies such as CIEL. Within the facility EHRs, only 50.5% (SD = 35.4%, p< 0.001) of patients had the nationally-

**Data Availability Statement:** All relevant data are within the paper and its Supporting Information files.

**Funding:** This work was supported in part by the NORHED program (Norad: Project QZA-0484). The

content is solely the responsibility of the authors and does not necessarily represent the official views of the Norwegian Agency for Development Cooperation. The funders had no role in study design, data collection and analysis, decision to publish, or preparation of the manuscript.

**Competing interests:** The authors have declared no competing interests exist.

endorsed patient identifier number recorded. Multiple regression analysis indicated the need for improvement on the mode of EHRs use of implementation.

## Conclusion

The standard EHRs use indicators can effectively measure EHRs use and consequently determine success of the EHRs implementations. The results suggest that most of the EHRs use areas assessed need improvement, especially in relation to active usage of the system and data exchange readiness.

## Introduction

Electronic Health Records systems (EHRs) have been introduced widely into medical processes in many countries worldwide, making patient data readily available for treatment, care and analysis [1–3]. These EHRs implementations promise to improve quality of patient care, patient safety and to reduce costs [4–6]. For instance, introduction of Electronic Medical records systems (EMRs) in health care has shown improvement in time dependent events such as patient waiting time, time to processing specimen in the laboratory from test request to results reporting among others benefits [7,8]. Moreover, a systematic review on utilization of EHRs for public health in Asia revealed their ability to help identify and predict seasonal outbreaks and high risk areas and prevent infections or diseases, leading to better health outcomes [9]. Schoen *et al.* noted an overall increase in EHR adoption and a significant variation in the growth rate across countries in their survey of primary care doctors in health reforms [10]. Despite the infrastructural and technical challenges experienced and reported in developing countries, the uptake of EHRs in healthcare processes have also been on the rise [2,11]. However, adoption of EHRs in Sub-Saharan Africa are largely driven by HIV treatment international programs such as President's Emergency Plan for AIDS Relief (PEPFAR) to support patient data management [11,12].

EHRs implementations involve a significant up-front investment in software design and development, infrastructure, implementation, training and IT support [13]. Sponsors, donors and management are demanding demonstrated value of EHRs implementations to inform investments and sustainability of the implementations [14,15]. Furthermore, EHRs implementations are complex, multi-faceted and impact healthcare organizations on many levels [15,16]. Consequently, chances of dismal performance of these systems are high, which may be unknown especially in public healthcare facilities. Therefore, it becomes necessary to evaluate information systems to provide evidence on system functional status and its fitness for purpose with a view to inform future deployments. Maximum benefits of information systems (IS) implementation can only be realized if the systems are deeply used in the post-adoption phase [17]. As such, evaluation of actual use of EHRs once implemented provides vital information relevant to informing approaches to improve success of existing and subsequent implementations.

Assessment of information system (IS) implementation success is both complex and never a straightforward task [18]. Thus, a range of evaluation methodologies and frameworks have emerged with divergent approaches, strengths, and limitations [19,20]. DeLone & McLean (D&M) IS success model is a mature and validated model for measuring health information systems success that was established in 1992 and revised in 2003 [21]. The model has been used to evaluate implementation success for a wide range of health information systems. Berhe

*et al.*, recently used the model to evaluate EMRs effectiveness from a user's perspective in Ayder Referral hospital in Ethiopia [22]. Cho *et al.* also used the model to evaluate the performance of newly-developed information systems in three public hospitals in Korea [23].

The revised D&M model has seven dimensions used to measure IS implementation success, namely: *System quality*, *Information quality*, *Service quality*, *System Use*, *intention to use*, *User satisfaction* and *Net benefits*. Of these dimensions, *'System Use'* was identified as the most appropriate variable for measuring the success of IS [21,24]. *System use* is the utilization of an IS in work processes by individuals, groups or organizations [11]. A number of studies have measured the actual EHRs use in terms of extent, frequency, duration of use and functions of the system based majorly on behavioural response of users through questionnaires, interview and/or focus group discussions [2,11,17,25,26]. However, only limited evaluation studies utilizing computer-generated data to assess EHRs use are available. This study was conducted to fill this gap by evaluating actual use of a national level EHR system implemented in healthcare facilities in Kenya, as a demonstration of how similar approaches could be applied across other low- and middle-income countries (LMICs) to evaluate use.

In most LMICs, measure of success of EHRs scale-up often relies on simple counts of the number of EHRs implementations. This study demonstrates that: (a) through use of standardized indicators [27], key new insights and gaps on actual status of EHRs implementations within countries use can be identified; (b) aspects of national-level EHRs usage assessments need not be time- or resource-intensive, as assessments can be automated using data already within the EHRs; and (c) mechanisms that allow efficient EHRs usage assessments offer insights to enable any identified EHRs usage gaps to be addressed in a timely manner.

## Materials and methods

### Study setting

This evaluation was conducted in Kenya, a country in East Africa with approximately 50 million persons [28]. Recognizing the role that EHRs play in patient data management, the government of Kenya through the Ministry of Health (MoH) and in collaboration with its development partners, namely Centres of Disease Control (CDC) and United States Agency for International Development (USAID), has implemented EHRs in over 1,000 public health facilities countrywide [29]. These implementations mainly support HIV care and treatment programs. While two EHRs (KenyaEMR and IQCare) by different vendors were initially endorsed for national deployment in support of HIV care, the country has since 2019 transitioned to supporting and deploying only KenyaEMR system (KeEMRs). In Kenya, KeEMRs is implemented in facilities spread across 22 Counties with varying numbers of sites per county (**S1 Appendix**). This study evaluated the actual use of KeEMRs within the facilities in which the system is deployed to inform actual EHRs usage across the country, based on computer-generated data. The study was conducted using census method with all 376 facilities that had KeEMRs implemented between 2012–2019 eligible to participate. For efficiency in care delivery, these public facilities are organised into Kenya Essential Package for Health (KEPH) service levels as follows: Level 1—community level; Level 2—dispensaries and clinics; Level 3—Health centres, maternity homes and sub district hospitals; Level 4—primary facilities which include District hospitals; Level 5—secondary facilities/Provincial hospitals; and Level 6—Tertiary/National hospitals.

### EHR system

KeEMRs is an implementation and adaptation of the open source OpenMRS system platform, which is widely deployed in many countries in Africa [30]. KeEMRs supports both

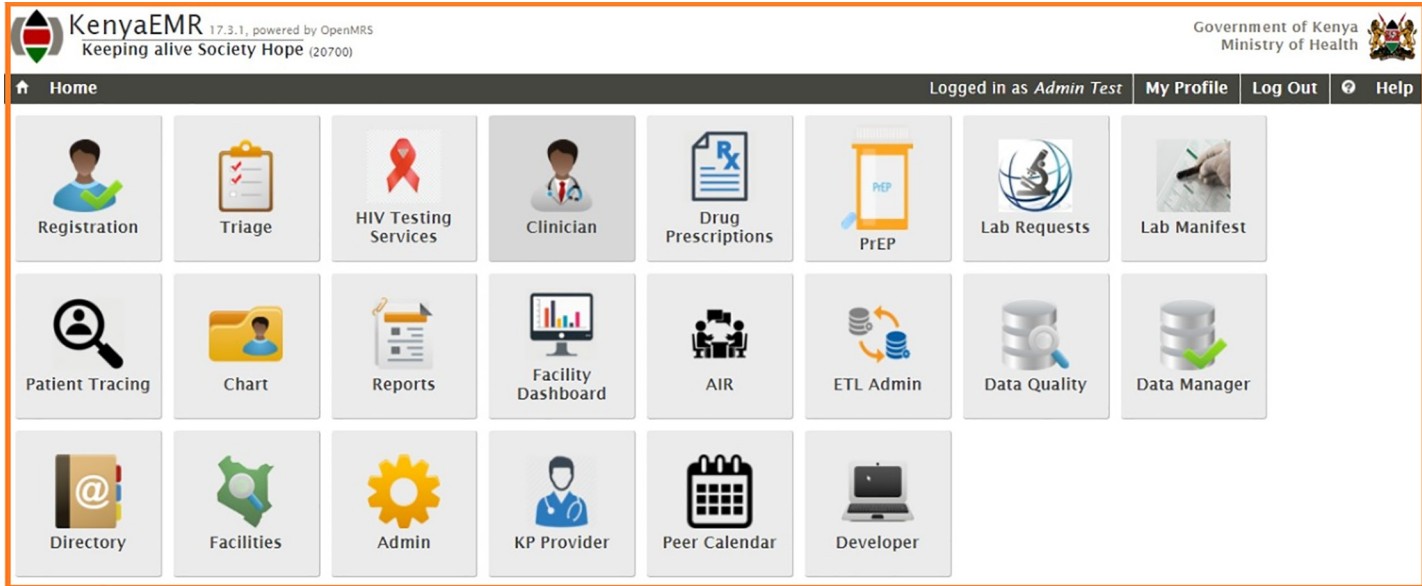

**Fig 1. Screenshot of KeEMRs home page.** Reprinted from [33] under a CC BY license, with permission from The Palladium Group- KeHMIS II Project, original copyright 2012.

retrospective and point-of-care data entry (RDE & POC) with most of the facilities equipped for POC implementation. It was designed and developed(customized) by International Training and Education Center for Health (I-TECH) in the year 2012 to support care and treatment of HIV/AIDS [31]. Currently, Kenya Health Management Information system II (KeHMIS II) project supports the implementation of KeEMRs in over 370 health facilities throughout Kenya [32]. **Fig 1** shows the homepage of the EHRs under study.

KeEMRs uses a communication layer referred to as interoperability layer (IL) to enable health data exchange with other health information systems such as pharmacy system (ADT). KeEMRs version 16.0.2 and above enforced the use of a nationally-endorsed 10-digit patient identifier number (five digits representing master facility list (MFL) code and five digits comprehensive care clinic number (CCCNo)) as from the year 2017 for unique patient identification.

## EHRs usage indicators

The EHRs use indicators used for this study are detailed in Ngugi et al [27]. The 15 rigorously derived indicators are modelled after the HIV Monitor, Evaluation and Reporting (MER) indicators, that facilities and implementations providing HIV care would be well-familiar with [34]. This study specifically focussed on the subset of the indicators that could be generated from within the implemented EHRs. This was because the ultimate goal is to have a module within the EHRs that can automatically generate indicators without human input for reporting and sharing with relevant stakeholders. The subset of the seven EHRs indicators included are outlined in **Table 1**. Three of the eight excluded indicators (namely *Reporting rate*, *Report timeliness* and *Report completeness*) rely on data in the national data aggregate system, the Kenya Health Information System (KHIS), and had already been evaluated and reported in a different study [29]. The other five excluded indicators (namely: *Data entry statistics*, *System Uptime*, *EHR Variable concordance*, *Report Concordance* and *Clinical Data timeliness*) required a level of human input to generate based on how the indicators are defined [27].

**Table 1. EHRs usage indicators evaluated.**

| # | Indicator (variable) | Domain | Indicator Measure | Indicator query description | Source of data |
|---|---|---|---|---|---|
| 1. | Staff system use | System use | Percentage of facility staff members who used the EHRs during the reporting period. | Defined by create, update, and delete actions around a patient record by an authorized EHRs user | EHRs |
| 2. | Observations (Clinical Volume) | System use | Number of mandatory HIV-related clinical data elements recorded for patients in the EHRs during the reporting period. | A count of the data captured by the 23 data elements* per patient encounter per month | EHRs |
| 3 | EHR Variable completeness | Data quality | The extent to which all required data elements for a patient are contained within the EHRs | No query. Data elements* captured from RDQA report generated from EHRs | EHRs |
| 4 | Data Exchange | Interoperability | Percentage of specified systems with which the EHRs can automatically exchange all required data with. | Count of unique data exchange messages between EHRs and other sub-systems through IL | EHRs |
| 5 | Standardized Terminologies | Interoperability | The proportion of key terminologies that are mapped to standard terminology services or use a nationally endorsed concept dictionary | % mapping of EHRs concepts with the concepts_reference_map table | EHRs |
| 6 | Patient Identification | Interoperability | Use of a nationally accepted patient identification method. | Patient visits identified using 10-digit identifier vs total active patients during the reporting period | EHRs |
| 7 | Automatic Reports | Reporting | The proportion of expected reports and sub-reports to the national level that are automatically generated and transmitted to the national reporting system. | A count of the reports' generation requests | EHRs |

*The 23 data elements include: Patient ID, sex, date of birth, date confirmed positive, enrolment date, initiation date, initial regimen, Current regimen, BMI at last visit date, TB screening at last visit, TB screening outcomes, IPT start date, IPT status, IPT outcome date, Second last VL result, second last VL date, most recent VL result, most recent VL date, last clinical encounter date, next visit date, Pregnancy assessment last date, Initial EID within 8 weeks, Infant prophylaxis.

## EHRs indicators queries

Queries were developed using MySQL to generate monthly indicator reports for the evaluated indicators except *EHR Variable Completeness*. These queries were programmed to be run within each EHRs implementation and were tested for accuracy in a training server prior to deployment. The data generated from the testing phase were reviewed by the researchers together with a data analyst to ensure validity of the indicator outputs (data) and needed revisions made to the queries. The resulting six queries were then combined and packaged into a script that comprised the queries and Linux bash script for creating a zipped archive file as an output after running the script. Pilot testing of the script was conducted in six facilities selected randomly in two counties to ensure feasibility of data collection within facilities. The final script was distributed to the study healthcare facilities, with accompanying instructions detailing the step-by-step process (**S2 Appendix**) for executing the script. Data for the *EHR Variable Completeness* indicator (key data elements related to HIV care and treatment) were derived from routine data quality assessment (RDQA) report that were already being generated from the EHRs.

## Data collection and analysis

All the 376 facilities implemented with KeEMRs were approached to participate in the study. Nevertheless, data collection script was distributed to 312 sites that gave authority for the commencement of the study and had used the EHRs for at least six months. Experienced system champions at each facility ran the query script as per the outlined protocol (**S2 Appendix**). Further support on running the query and generating the report was provided through a toll-free line to the EHRs developers helpdesk as needed. Monthly indicator data were generated from each EHRs implementing facility from January 2012 (the earliest possible time for system

deployment) to December 2019. Generated reports (data) were transmitted electronically to the research team for consolidation and data cleaning thereby enforcing data quality. No personal identifiable information were contained in the resulting indicator reports. All the EHRs implementations used the same terminology service, hence assessment of the *Standardized Terminologies* indicator evaluated the proportion of terms in this dictionary that mapped to standard terminologies such as SNOMED and ICD [35,36]. Data collection for this study occurred over a period of eight weeks between April and June 2020.

Facility characteristics (KEPH levels, facility-type-category, ownership, services and mode of EHRs use) data were derived from Master Facility List (MFL) website maintained by the MoH. These data were summarized using descriptive statistics. Mean values and standard deviations of the collective performance by facilities for each indicator were calculated. One-way ANOVA (with Tukey's b "post-hoc" test) were performed to measure the variance in variables means (*Staff System Use*, *clinical volume*, and *patient identification* indicators) across the counties. Correlation analysis was also performed to measure the relationship between *staff system use* indicator and volume of the clinical data for insight on user productivity. Weighted mean of *Staff System Use* and *Patient Identification* indicators was computed to determine the overall performance of each facility. The two indicators assumed a weighting mean of 1, hence each was assigned a weight of 0.5 in order to have an unbiased mean. A summation of the weighted mean of the two indicators for each facility was then computed and finally ranked in descending order. The two indicators were chosen because they are the key variables that show EHRs utilization in the facility. Data exchange indicator data were treated and analysed as dichotomous data (presence or absence) of interoperability layer (IL) software that facilitates data exchange with external systems.

Finally, we fitted multiple linear regression model to establish how individual facility characteristics affected the use of the system. The dependent variable was number of active system users while the covariates were the facility characteristics (KEPH level, ownership services and mode of EHRs use). All analyses were performed using IBM SPSS statistics 25 [37].

The primary outcome of interest for this study was to determine the collective performance by facilities on each of the seven indicators over the period of KeEMRs implementation in Kenya, as a measure of overall EHRs usage. In addition, this study had several secondary outcomes of interests, namely: (a) evaluation of variability in EHRs usage between counties, (b) relationship between number of active users of systems and the clinical volume for insight on user productivity, and (c) the effect of facility characteristics on EHRs use.

### Ethical statement

The study was approved by the Institutional Review and Ethics Committee at Moi University, Eldoret (MU/MTRH-IREC approval Number FAN: 0003348). Permission to collect data was also obtained from Ministry of Health (MoH), County Directors of Health of each county, as well as Service Delivery Partners (SDPs) responsible for EHRs implementations and HIV data at the facility level. Permission to collect data from 312 (out of 376) facilities in 19 counties were granted. All participants filled a consent form before taking part in the study. No personal identification data were collected from either patient records/system database or the healthcare facilities or personnel who executed the queries.

## Results

### Organizational characteristics of the responding facilities

Out of the 312 facilities that assented to participate in the study, 213 (68.3%) spanning 19 Counties responded. Characteristics of the responding facilities are detailed in **Table 2**. The

**Table 2. Frequency distribution for the facility characteristics (n = 213).**

| Characteristics | Count | % | P-value |
|---|---|---|---|
| **KEPH Level** | | | |
| Level 2 | 28 | 13.10% | 0.092 |
| Level 3 | 100 | 46.90% | |
| Level 4 | 85 | 39.90% | |
| Total | 213 | 100.00% | |
| **Facility type category** | | | |
| Dispensary | 26 | 12.20% | 0.057 |
| Health Centre | 99 | 46.50% | |
| Hospitals | 86 | 40.40% | |
| Medical Clinic | 2 | 0.90% | |
| Total | 213 | 100.00% | |
| **Ownership** | | | |
| Faith Based Organizations | 21 | 9.90% | 0.001 |
| Ministry of Health | 189 | 88.70% | |
| Non-Governmental Organizations Private | 3 | 1.40% | |
| Total | 213 | 100.00% | |
| **Services** | | | |
| CT* | 161 | 72.30% | <0.001 |
| CT&HTS** | 52 | 13.60% | |
| Total | 213 | 100.00% | |
| **Mode of use** | | | |
| HYBRID | 112 | 52.60% | <0.001 |
| POC | 20 | 9.40% | |
| RDE | 81 | 38.00% | |
| Total | 213 | 100.00% | |

* Care & Treatment service (CT)

**HTS–HIV counselling and testing service.

responding facilities were largely between KEPH levels 2 and 4, as these were the ones offering HIV services and in which the EHRs was deployed. Most of these facilities offered care and treatment (C&T) service 161(72.3%). Over 86% of these facilities were either Health Centres or Hospitals and were largely owned and run by the Ministry of Health (88.7%). Only 9.4% of the facilities were completely paperless, with slightly over a third of the facilities (38.0%) still doing retrospective data entry (RDE) fully.

The total number of responding facilities with EHRs implementation varied by county, with the lowest county having three while the highest had 25. Most of these implementations occurred in 2014 (113 implementations, 53.1%) followed by 2013 (91 implementations, 42.7%) (**S3 Appendix**). No implementations occurred in the period 2015–2017 whilst there were only four new implementations (1.8%) between 2018 and 2019 in line with the country's planned implementation strategy.

## EHRs usage indicator results

**Staff system use.** An average of 18.1% (SD = 13.1%) staff members with EMRs access rights used the system in any given period. The best and worst facilities had a mean usage of 46.8% (SD = 23.3%) and 7.3% (SD = 3.3%) respectively (p< .001) (**S4 Appendix**).

**Observations (clinical data volume).** On average, the facilities captured 3,363 (SD = 4,249) patient-related data elements (clinical data volume) monthly, based on the mandatory 23 data types of interest for HIV reporting in Kenya [38] showing there was high dispersion in the data collected (**S4 Appendix**). The facility with highest mean monthly volume captured 28,937 (SD = 11,356) data elements while the least had 251 (SD = 167). There was a weak positive correlation between *Observations* (*Clinical data volume*) and *Staff System Use* indicators (coefficient r = 0.01).

**EHR variable completeness.** We observed that all the 23 data elements required for HIV patients by the MoH were contained within the records for each patient in the EHRs. Hence the *EHR Variable Completeness* indicator as per the country's standard operating procedures (SOP) was 100% across the study facilities.

**Data exchange.** Majority of the facilities (183/213) lack the interoperability layer (IL) module and hence had no capability to exchange health data with external systems (**S5 Appendix**). Of the 14.1% facilities which had data exchange capability, 56.7% of them were in one county. None of the facilities (n = 108) in 13 of the 19 counties had data exchange capability.

**Standardized terminologies.** On average 97.6% (52,098 out of 53,353) of KeEMR system concepts were mapped to the standardized (international) terminologies/concept dictionaries such as CIEL and SNOMED.

**Patient identification.** Only 50.5% (SD = 35.4%, p< 0.001) of the patient records had patients with identifiers in the nationally-endorsed patient identifier format (10-digit number = 5 MFL+5 CCCNo.) (**S4 Appendix**). There was a wide range of 3% to 100% conformity across the facilities, indicating the need for further investigation on why such low conformity rates. Three of the healthcare facilities fully adopted the approved patient identifier (100%) while 28 facilities had an average mean of < 10% conformity in the use of the national patient identifier.

**Automatic reports.** KeEMRs is configured to generate monthly Ministry of Health routine reports (MoH 731) for transmission to the national reporting system (KHIS). However, by the time of this study, we could not capture the data to compute automatic reports indicator (the proportion of expected reports to the national level that are automatically generated and transmitted to the national reporting system). This was because the records of the generated reports and their transmission are not saved, with tables refreshed on a daily basis.

## Performance of the facilities

Using the weighted mean of the means scores of *Staff System Use* and *Patient Identification* indicators, facilities were benchmarked against each other using the "best performer" and "worst performer" approach. The weighted mean ranged from 9% to 65% across the 213 facilities. **S6 Appendix** presents facility performance list from the highest to the lowest. The top ten performing facilities had an average weighted mean of 61% (range 59–65%) while the bottom ten facilities had an average mean of 11% (range 9–12%).

## EHRs use against facility characteristics

The relationship between the facility characteristics and the number of active system users assessed by the multiple linear regression analysis was statistically significant (p = 0.000) for all the covariates (**Table 3**). The characteristics influenced system usage positively, with the exception of Mode of EHRs use characteristic. RDE mode of EHRs use had the highest negative impact on the use of the system.

**Table 3. Multiple linear regression model for staff system use and facility characteristics.**

| Facility Characteristics | | Unstandardized Coefficients | | Standardized Coefficients | t | P-value |
|---|---|---|---|---|---|---|
| | | B | Std. Error | Beta | | |
| (Constant) | | 0.354 | 0.084 | | 4.213 | 0.000 |
| KEPH Level | Level 2<br>Level 3<br>Level 4 | 0.445 | 0.019 | 0.194 | 23.929 | 0.000 |
| Ownership | -Faith-Based Organisation<br>-Ministry of Heath<br>-Non-Governmental Organization | 0.401 | 0.035 | 0.092 | 11.308 | 0.000 |
| Services | CT<br>CT&HTS | 0.392 | 0.015 | 0.206 | 25.351 | 0.000 |
| Mode of EHRs use | Hybrid<br>POC<br>RDE | -0.124 | 0.014 | -0.074 | -9.176 | 0.000 |

Dependent Variable: Number of active system users; Independent Variables: KEPH level, ownership, mode EHRs of use, and services. p-value: When p< = 0.05, there is statistically significant difference. B (coefficient) explains a change in dependent variable that can be attributed to a change of one unit in the independent variable.

## Discussion

To our knowledge, this is the first national-level study that has systematically evaluated actual EHRs use post-implementation utilizing computer-generated real-time data based on robustly developed EHRs usage indicators. A systematic review on measuring EHRs use in primary care revealed that most studies measured use through assessing the utilization of individual EHRs functions [26]. The findings from our study highlight the fact that simply counting number of EHRs implementations is highly inadequate in determining IS implementation success. Multidimensional set of indicators for evaluating EHRs use in this study align with the three main components of EHRs meaningful use, namely: (1) EHRs must be used in the care processes such as prescribing, (2) EHRs must encompass electronic health data exchange for improved health care quality and (3) EHRs must support reporting of clinical measures [39,40]. In this study, indicators reflecting system use and interoperability domains indicated low measures, suggesting the need for further improvement.

Measuring system-use at the application level sheds light on how fully or effectively organizations are using IT [41]. In our study, the overall *Staff System Use* was very low across all the facilities regardless of the period of EHRs implementation. The study established existence of many dormant user accounts in the EHRs across all facilities portraying a high number of users authorized to use the system (denominator) compared to actual number of users (numerator) hence the low average mean. Another possibility of low mean could have been occasioned by shared login credentials or shared computers resulting into multiple users on one user account. This presented a scenario like only one user performed activities around patients' files i.e., create, update or delete, which were the assessed *Staff system use* indicator measures. Consequently, this compromised the accuracy of the numerator count. A study investigating users' behaviour in password utilization revealed users share passwords for convenience as well as a show of trust [42]. The finding from our study warrants deeper assessment on user credentialing processes and account usage patterns (such as sharing of credentials). It also highlights the need to re-emphasize good password practices to the system users and active monitoring of user accounts by the system administrators. We also recommend further research to establish user-computer ratio in the healthcare facilities.

While our results show KeEMRs' readiness to interoperate with other external systems due to high mapping rate of its concepts to standard terminology services like CIEL [38,43], the study established a slow incorporation of the interoperability layer (IL) within the EHRs. Integration with other systems is one of system quality measures among ease-of-use, functionality, reliability and flexibility [44]. The low data exchange indicator findings from this study suggests the need for investigation on other system quality measures. Technological barriers, such as functionality and compatibility issues, and non-user-friendliness can limit system use [45]. The actual uptake of the nationally-accepted patient identifiers was average although with large variations in uptake levels between facilities and between counties. Several studies reveal lack of interoperability as a well-known impediment to EHRs successful adoption and use [46–49]. As such, interoperability layer should be incorporated into all EHRs implementations as well as concerted efforts towards nationwide adoption and use of unique patient identifier, which promises to improve patient safety and care efficiency [50].

The study expected a strong positive correlation between *Staff System Use* and *Observations (*clinical data volume) recorded in the EHRs, which was not the case. This could be attributed to the possibility of users sharing login credentials as intimated earlier. Several factors determine facility clinical volume such as patients' volume, frequency of patients' visits (encounters), EHRs mode of use and active usage of the system during care, all unique to each facility. Ideally, facilities entering data retrospectively should efficiently transfer paper records into the EHRs in a timely fashion for 100% concordance. However, a study on EHRs use and user satisfaction by Tilahum and Fritz revealed retrospective data entry as a major cause of dissatisfaction of EHRs use among users, especially when the same individuals collecting the data are tasked to enter it into the system later [2]. Indeed, our study revealed that point of care (POC) and hybrid modes of data capture were associated with increased system usage compared to retrospective data entry. Thus, EHRs implementors should aim at point of care mode of operation right from initiation.

## Study strengths and limitations

The key strength of the study was the use of empirical data extracted directly from EHRs hence not subject to bias normally introduced by human judgment prevalent in self-reports such as questionnaires. Boon *et al* in their study on antecedents of continued use and extended use of enterprise systems strongly recommended use of system log file data to overcome human related response bias [51]. Secondly, the study period (2012–2019) was long enough to reveal the state of the EHRs use in the health care facilities. Also, the study results are reliable due to the use of census method in the collection of the primary data. Furthermore, these facilities had diverse range of characteristics in terms of ownership and facility levels and covered broad geographic area of Kenya. The study does, however, acknowledge a few limitations. It was only conducted in one country (Kenya) and the findings do not necessarily translate directly to other countries. However, the study provides a demonstration case that can be modelled by other countries to inform similar EHRs usage evaluations. Finally, this study only focused on facilities where the EHRs were in actual use, without mention of locations where the EHRs were implemented and actually failed. Attention needs to be paid to failed implementations, to ensure that usage rates are not being over-reported.

In the next step of our research, we will conduct qualitative assessments to better understand the observed findings. This will be done through Focus Group Discussions (FGD) and semi-structured interviews with EHRs users and key stakeholders. Further, we will work with relevant partners to help integrate outputs and visualizations of the usage reports within the EHRs, and to provide various visualizations and dashboards for managers and decision-

makers to increase visibility on system usage within and across facilities. It is also recognized that continued usage of EHRs in the patient care processes do not necessarily lead to better work performance or improved care quality. Further research is needed to investigate impact of EHRs usage on care quality and outcomes.

## Conclusion

Assessment of actual use of implemented EHRs within LMICs is important. The systematically generated standard EHRs usage indicators can be adopted and used successfully within facilities across countries. Results from this study demonstrate that there are many areas of improvement in EHRs use, as well as the need for continuous monitoring of EHRs use to inform timely interventions. Simply counting number of implementations, as is currently done in many settings, remains a highly inadequate measure for evaluating EHRs implementations success.

## Supporting information

**S1 Appendix. Distribution of KeEMRs implementations as of June 2020.**
(PDF)

**S2 Appendix. Standard operating procedures for query extraction.**
(PDF)

**S3 Appendix. KeEMRs implementations distribution in the period 2012–2019 across the counties (n = 19).**
(PDF)

**S4 Appendix. Facilities descriptive statistics for staff system use, observations & patient identification indicators.**
(XLSX)

**S5 Appendix. Interoperability layer (IL) module (data exchange) presence/absence in facilities across the counties.**
(PDF)

**S6 Appendix. Facilities performance using weighted means.**
(XLSX)

**S7 Appendix.**
(XLSX)

## Acknowledgments

Authors would like to acknowledge KeEMR system developers for providing input into the testing of the study instrument (query script), and helpdesk support to study participants. We also appreciate logistical support by Kenya Ministry of Health, County health directors, AMPATH Plus and FACES service development partners, and County Health Records Information Officers (CHRIOs). Much appreciation also to all the healthcare facilities and the system champions for their participation in the study.

## Author Contributions

**Conceptualization:** Philomena Ngugi, Martin C. Were.

**Data curation:** Philomena Ngugi.

**Formal analysis:** Philomena Ngugi.

**Funding acquisition:** Martin C. Were.

**Investigation:** Philomena Ngugi.

**Methodology:** Philomena Ngugi, Ankica Babic, Martin C. Were.

**Project administration:** Ankica Babic, Martin C. Were.

**Resources:** Martin C. Were.

**Software:** Philomena Ngugi.

**Supervision:** Ankica Babic, Martin C. Were.

**Validation:** Philomena Ngugi.

**Writing – original draft:** Philomena Ngugi, Ankica Babic, Martin C. Were.

**Writing – review & editing:** Philomena Ngugi, Ankica Babic, Martin C. Were.

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
