## [Decision Letter · Decision Letter 0]

30 Mar 2021

PONE-D-21-01911

A multivariate statistical evaluation of actual use of electronic health records systems implementations in Kenya

PLOS ONE

Dear Dr. Ngugi,

Thank you for submitting your manuscript to PLOS ONE. After careful consideration, we feel that it has merit but does not fully meet PLOS ONE’s publication criteria as it currently stands. Therefore, we invite you to submit a revised version of the manuscript that addresses the points raised during the review process.

We look forward to receiving your revised manuscript.

Kind regards,

Chaisiri Angkurawaranon

Academic Editor

PLOS ONE

Journal Requirements:

2. We note that S2 Appendix in your submission contains map images which may be copyrighted.

We require you to either (a) present written permission from the copyright holder to publish this figure specifically under the CC BY 4.0 license, or (b) remove the figure from your submission:

a. You may seek permission from the original copyright holder of S2 Appendix to publish the content specifically under the CC BY 4.0 license. 

b. If you are unable to obtain permission from the original copyright holder to publish this figure under the CC BY 4.0 license or if the copyright holder’s requirements are incompatible with the CC BY 4.0 license, please either i) remove the figure or ii) supply a replacement figure that complies with the CC BY 4.0 license. Please check copyright information on all replacement figures and update the figure caption with source information. If applicable, please specify in the figure caption text when a figure is similar but not identical to the original image and is therefore for illustrative purposes only.

'Authors would like to acknowledge KeEMR system developers for providing input into the testing of the study instrument (query script), and helpdesk support to study participants. We also appreciate the support by Kenya Ministry of Health, County health directors, AMPATH Plus and FACES service development partners, and County Health Records Information Officers (CHRIOs). Much appreciation also to all the healthcare facilities and the system champions for their participation in the study.'

'This work was supported in part by the NORHED program (Norad: Project QZA-0484). The content is solely the responsibility of the authors and does not necessarily represent the official views of the Norwegian Agency for Development Cooperation.  The funders had no role in study design, data collection and analysis, decision to publish, or preparation of the manuscript.'

Please provide an amended statement that declares *all* the funding or sources of support (whether external or internal to your organization) received during this study, as detailed online in our guide for authors at http://journals.plos.org/plosone/s/submit-now 

Please also include the statement “There was no additional external funding received for this study.” in your updated Funding Statement.

4. We note that Figure 1 in your submission contains copyrighted images.

All PLOS content is published under the Creative Commons Attribution License (CC BY 4.0), which means that the manuscript, images, and Supporting Information files will be freely available online, and any third party is permitted to access, download, copy, distribute, and use these materials in any way, even commercially, with proper attribution. For more information, see our copyright guidelines: http://journals.plos.org/plosone/s/licenses-and-copyright.

We require you to either (a) present written permission from the copyright holder to publish this figure specifically under the CC BY 4.0 license, or (b) remove the figure from your submission:

b. If you are unable to obtain permission from the original copyright holder to publish this figure under the CC BY 4.0 license or if the copyright holder’s requirements are incompatible with the CC BY 4.0 license, please either i) remove the figure or ii) supply a replacement figure that complies with the CC BY 4.0 license. Please check copyright information on all replacement figures and update the figure caption with source information. If applicable, please specify in the figure caption text when a figure is similar but not identical to the original image and is therefore for illustrative purposes only.

Reviewers' comments:

Reviewer's Responses to Questions

**Comments to the Author**

1. Is the manuscript technically sound, and do the data support the conclusions?

Reviewer #1: Partly

Reviewer #2: Partly

2. Has the statistical analysis been performed appropriately and rigorously? 

Reviewer #1: I Don't Know

Reviewer #2: Yes

3. Have the authors made all data underlying the findings in their manuscript fully available?

Reviewer #1: No

Reviewer #2: Yes

4. Is the manuscript presented in an intelligible fashion and written in standard English?

Reviewer #1: Yes

Reviewer #2: Yes

5. Review Comments to the Author

Reviewer #1: I read the paper with interest and I can see that there is some interesting data here. However I was not really convinced by some aspects of the work as presented here and found the paper rather too long and a bit repetitious while not really focusing on a clear set of issues and a strong message for the reader.

In the introductory section a number of older references are used to set the scene. The paper would be stronger if it focused on more recent studies in its literature review and on literature that serves the needs of this particular study e.g. that details specific metrics of system use, , assessment of system value, and the analysis of log files. Some of what was here felt a bit out-of-date (references over 10 years old). It hardly mentioned core models of system usage and success e.g. DeLone and Mclean get a very brief mention but their model – first and revised version – may be very useful to underpin this work.

I would be more interested in the data that is collected here if it was not described (on a couple of occasions) as ‘objective’ or ‘un biased’. From my position this data is useful, interesting, but it is not a higher truth than other data collected in other ways. Nor is it (or any other data) un-biased – computer systems and their data are biased from the day they are developed! I would like more discussion of the validity and potential of the data and independent variables, and how they might be interpreted or add value. But this cannot be asserted or taken for granted..

Similarly I felt that the focus on the GLM model was the weakest part of the paper. To reduce the complex reality of clinical work and use of EHR over years to a single model seems a rush to judgement. The discussion the Table 4’s findings seemed to be more description than an evaluation or analysis of the results (although the discussion section did take this a little further – e.g. page 17).

Before any such model I would suggest more attention paid to the data itself and to the various correlations that it reveals.

I was not clear what the section on ‘Implementation by county (page 11 and 12) added to the paper – just a large table . The data on implementation year is interesting but could find a home elsewhere in the paper

I was puzzled by the definition of some variables. For example, is the ‘EHR Variable completeness’ (page 13) really 100%in any meaningful way? Is the fact that then data fields are defined in the software really enough to say this? A real measure might be the number of these fields that are actually in use – e.g. populated in over x% of patient records?

I was also puzzled by the Observations measures (page 13). I would have thought this would need a denominator to of any interest – and what the denominator should be would be (patient encounters, clinical staff-users etc) is an interesting issue to address in the paper at some depth, but it only appears briefly in the discussion (page 17).

In summary I believe that the paper could be significantly improved by tightening the presentation and some of the repetition of findings, focusing on interpreting the data more and less on the GLM. I would also suggest a stronger discussion section that takes the reader further and focuses on the ‘so-what ‘. The present version seems to repeat the main findings, but not take the argument further.

Reviewer #2: Thank you for the opportunity to review this manuscript on actual use of electronic health records systems. Since the advent of electronic health records, there have been concerns about the benefits of using EHRs and the success of implementation of EHRs. In the research conducted, the authors examined seven indicators that might reflect the actual use of electronic health records.

Upon reading the manuscript, the question arises whether the authors are really identifying the actual use of EHRs by health care staff with the seven indicators they have chosen.

The first indicator is ‘staff system use’ and was measured by the percentage of facility staff members who used the EHRs during the reporting period. In their study, the authors found a low average of 18.2%. However, this number has little meaning without providing more information about the context. For instance, are the health care facilities using a few stationary computers or are they using a shared or individual iPad? With a stationary computer health care professionals often log in with one user account and subsequently use the EHRs with multiple professionals on one user account. This is also the case when working with a shared iPad. Therefore, the ‘staff system use’ as measured by the authors is not an accurate indicator for actual use of the EHRs.

In addition, another indicator is ‘standardized terminologies’. This indicator is measured through the proportion of key terminologies that are mapped to standard terminologies. Mapping of terminologies is mostly relevant for the exchange of health care information and for the use of routine care data for research purposes. However, this mapping has no direct influence on the use of EHRs by health care staff in daily care practice. So, how does this indicator say anything about the actual use of EHRs? That remains unclear in this study. Moreover, there are different standardized terminologies for doctors, nursing staff and other health care staff. The authors lack to explain by which health care staff the EHRs are being used. Therefore, the indicator standardized terminology remains vague and unclear.

Besides, the other five indicators can also be questioned whether they are an accurate indicator for actual use of the EHRs by health care staff. Therefore it is very doubtful whether the authors in this study really assessed the state of the implementations of EHRs.

The authors already mentioned themselves that the study findings are not generalizable to other countries. They suggest that their approach for analysing EHRs use can be generalizable to other countries. However as explained before, this approach seems very questionable. Therefore, the added value of this manuscript for the international research community is not clear.

More detailed comments:

- Use of abbreviations. The authors are not consistent in use of abbreviations for the electronic health records. They use both EHR, EHRs and EMRs which causes confusion. Besides, the authors use a lot of other abbreviations which is not beneficial for the readability of the manuscript. The authors should only use universally used abbreviations and only when necessary. This applies both to the body of text and the tables.

- In the ‘Introduction’ section the authors write that EHRs improve quality of care and support HIV programs at a national level. Authors should explain these suggested relations further, since it remains unclear.

- In the ‘Introduction’ section the IS success model is introduced without further describing the model. For readers who are not familiar with this model this paragraph is unclear. Further explanation is needed.

- In the ‘Material and Methods’ section the authors write there are two types of EHRs endorsed for national development. It is unclear what was meant by ‘two types’. Does this mean EHRs from two different vendors?

6. PLOS authors have the option to publish the peer review history of their article (what does this mean?). If published, this will include your full peer review and any attached files.

Reviewer #1: **Yes: **Tony Cornford

Reviewer #2: No

---

## [Author Response · Author response to Decision Letter 0]

14 May 2021

We have responded comprehensively all the issues raised by the reviewers

---

## [Decision Letter · Decision Letter 1]

29 Jun 2021

PONE-D-21-01911R1

A multivariate statistical evaluation of actual use of electronic health records systems implementations in Kenya

PLOS ONE

Dear Dr. Ngugi,

Thank you for submitting your manuscript to PLOS ONE. After careful consideration, we feel that it has merit but does not fully meet PLOS ONE’s publication criteria as it currently stands. Therefore, we invite you to submit a revised version of the manuscript that addresses the points raised during the review process.

We look forward to receiving your revised manuscript.

Kind regards,

Chaisiri Angkurawaranon

Academic Editor

PLOS ONE

Journal Requirements:

Reviewers' comments:

Reviewer's Responses to Questions

**Comments to the Author**

1. If the authors have adequately addressed your comments raised in a previous round of review and you feel that this manuscript is now acceptable for publication, you may indicate that here to bypass the “Comments to the Author” section, enter your conflict of interest statement in the “Confidential to Editor” section, and submit your "Accept" recommendation.

Reviewer #1: (No Response)

Reviewer #2: (No Response)

2. Is the manuscript technically sound, and do the data support the conclusions?

Reviewer #1: Yes

Reviewer #2: Yes

3. Has the statistical analysis been performed appropriately and rigorously? 

Reviewer #1: I Don't Know

Reviewer #2: I Don't Know

4. Have the authors made all data underlying the findings in their manuscript fully available?

Reviewer #1: Yes

Reviewer #2: Yes

5. Is the manuscript presented in an intelligible fashion and written in standard English?

Reviewer #1: Yes

Reviewer #2: Yes

6. Review Comments to the Author

Reviewer #1: I found this revised manuscript better in many respects,. It is tighter in its focus , less repetitious and has lost some extraneous material. I was however disappointed to see that the rebuttal letter’ was so brief and indicated so little about how the authors had chosen to revise the paper.

The way that the authors cite and use the DeLone and Mclean model is better in this version, but is still rather unimaginative. To say that the model has 6 dimensions and then only focus on one (‘system use’) is to underplay the model and the data available here. For example, is there not data on information quality here, and perhaps system quality too? Clearly this is not a ‘full’ D&M data set – but their work can be really helpful in interpreting and validating the data and analysis (more so than an add hoc GLM).

I am still confused as how the ‘Observations’ measure is to be interpreted without some denominator. As a raw value it seems to be a proxy for so many things (size, enthusiasm, patient type, resources etc. etc.) as to be of little use.

As I said in my earlier review, I am not convinced by the GLM analysis, and I am not sure that he findings here tell us very much. If the authors think otherwise then I encourage them to draw out the importance of these findings to convince me or other readers. - I note that in this version f he paper relatively little discussion is devoted to the GLM findings.

Overall, I believe that the paper does have interesting information to convey, and has potential for an analysis of this data that can make a contribution to the study of EHR in LMICs and in particular to studies of systems use. I do however suggest a further revision and edit to catch a number of small language issues and to strengthen the discussion of the data and its ability to reflect use. I would also tone down claims such as on page 18)to ‘conclusively give a true state of EHRs use’. That is a claim that no researcher across the world can make!! Equally I don’t think you can claim ‘highly reliable data’ (same page). I suggest fewer of such claims and more time addressing the subtlies of the data, set in the context. This will impress a reader far more.

Reviewer #2: Thank you for addressing the comments raised in a previous round of review and adjusting your manuscript. I believe the adjustments have strengthened the manuscript.

Yet, I have one remark left about the adjustments made regarding the indicator ‘staff system use’. It was good to read you now address the issue of sharing individual-passwords. However, you very quickly draw the conclusion that training on appropriate use of account credentials is needed for staff. Thereby you assume that a lack of knowledge is the underlying cause, yet how do you know such a lack of knowledge exists? Can you refer to studies that indicate such a lack of knowledge? Other studies often point towards problems with the user-friendliness of EHRs, instead of a lack of knowledge among staff. Therefore, I believe you should look again if you can substantiate your conclusion/recommendation.

7. PLOS authors have the option to publish the peer review history of their article (what does this mean?). If published, this will include your full peer review and any attached files.

Reviewer #1: **Yes: **Tony Cornford

Reviewer #2: No

---

## [Author Response · Author response to Decision Letter 1]

3 Aug 2021

Re: Manuscript PONE-D-21-01911R1: A multivariate statistical evaluation of actual use of electronic health record systems implementations in Kenya.

We appreciate the second review by PLOS ONE Journal of our Manuscript entitled “A multivariate statistical evaluation of actual use of electronic health record systems implementations in Kenya.” We are grateful for the opportunity to respond comprehensively to the reviewers’ comments. Please find our responses to all the comments by the reviewers below:

Editor’s Comments

Please review your reference list to ensure that it is complete and correct. If you have cited papers that have been retracted, please include the rationale for doing so in the manuscript text, or remove these references and replace them with relevant current references. Any changes to the reference list should be mentioned in the rebuttal letter that accompanies your revised manuscript. If you need to cite a retracted article, indicate the article’s retracted status in the References list and also include a citation and full reference for the retraction notice

Response: We confirm that we have reviewed our reference list and ensured that it is complete and correct. 

We have effected the changes on the reference list as follows:

i) Reference no.6 - Revised by adding details to the reference

ii) Reference no. 10 - Corrected name of author

iii) Reference 34 – corrected the reference details

iv) Reference 37 – removed from the list

v) Added references 41,42, 44 & 45

Reviewers’ Comments

Reviewer #1:

I found this revised manuscript better in many respects,. It is tighter in its focus , less repetitious and has lost some extraneous material. I was however disappointed to see that the rebuttal letter’ was so brief and indicated so little about how the authors had chosen to revise the paper.

Response: We really appreciate the positive feedback on the revisions of our manuscript. Your comments have strengthened our manuscript. We do sincerely apologize where we were too brief in explaining our approach in addressing the raised concerns. Our intent was to answer comprehensively and to the point.

1. The way that the authors cite and use the DeLone and Mclean model is better in this version, but is still rather unimaginative. To say that the model has 6 dimensions and then only focus on one (‘system use’) is to underplay the model and the data available here. For example, is there not data on information quality here, and perhaps system quality too? Clearly this is not a ‘full’ D&M data set – but their work can be really helpful in interpreting and validating the data and analysis (more so than an add hoc GLM).

Response: We appreciate reviewer’s feedback and also take note of the concerns/suggestions. The main study on which our manuscript is based was a summative evaluation to assess the success of EHRs implementations in healthcare facilities after eight years of rollout nationally at different stages (implementation periods). The choice of only one success variable in the DeLone & McLean IS success model, in this case ‘system use’, was informed by IS researchers argument that “no single variable is intrinsically better than another, so the choice of success variables is often a function of the objective of the study, the organizational context . . . etc.”(Page 17 [1]). We found ‘system use’ success variable very suitable for our study objective which was to establish actual use of the implemented EHRs. The study is also part of information system (IS) effectiveness success measure, evaluated by use, user satisfaction and net benefits variables of D&M model.

The data collected in our study is not clinical related but reveals the EHRs use patterns (e.g user activities in the system assessed by Staff system use indicator). Indicators assessing data quality (information quality) concerning whether the data in the EHRs are relevant, comprehensive, precise, and providing adequate overview of clinical work were not considered in this phase of the study. However, EHRs readiness to capture quality data was assessed by EHR variable completeness indicator (Page 12/13, lines 264-267). 

We have also strengthened the discussion part in relation to system quality and the discussion on the data as follows:

“Integration with other systems is one of system quality measures among ease-of-use, functionality, reliability and flexibility [43]. The low data exchange indicator findings from this study suggests the need for investigation on other system quality measures. Technological barriers, such as functionality and compatibility issues, and non-user-friendliness can limit system use [44].” (Page 16, lines 340-344)

2. I am still confused as how the ‘Observations’ measure is to be interpreted without some denominator. As a raw value it seems to be a proxy for so many things (size, enthusiasm, patient type, resources etc. etc.) as to be of little use. 

Response: We appreciate reviewer’s comments on the Observations indicator. We agree a denominator may well be of help in better interpretation of the data. However, as explained in our earlier rebuttal letter, in this study, we followed the approved usage indicators as defined in Ngugi et al. study (https://doi.org/10.1371/journal.pone.0244917). Therefore, we are limited by the description on how to collect the clinical data by what is described in the manuscript. However, revisions on the indicator to incorporate a denominator may be considered in a future study (or revision of the indicators). For example, the ‘observations’ measure data can be matched against the national data warehouse data to see if required observational data is sent to the data warehouse as defined (i.e. concordance). We hope that this explanation clarifies and is satisfactory regarding the observation indicator.

3. As I said in my earlier review, I am not convinced by the GLM analysis, and I am not sure that he findings here tell us very much. If the authors think otherwise then I encourage them to draw out the importance of these findings to convince me or other readers. - I note that in this version f he paper relatively little discussion is devoted to the GLM findings.

Response: We have taken note of the comments on GLM analysis and apologize for the confusion. We have replace the GLM analysis with multiple linear regression model to establish how individual facility characteristics (level of hospital, ownership, services offered and mode of EHRs use) affected the use of the system. The result from the analysis shows the units by which each facility characteristic influenced system use (positively or negatively). We believe this is vital information to the system implementers and the Ministries of health, informing areas that need attention to improve success of existing and subsequent implementations. 

We have revised the respective sections of the manuscript as follows:

Data analysis:

“Finally, we fitted multiple linear regression model to establish how individual facility characteristics affected the use of the system. The dependent variable was number of active system users while the covariates were the facility characteristics (KEPH level, ownership services and mode of EHRs use).” (Page 10, Lines 211-214)

Results:

“The relationship between the facility characteristics and the number of active system users assessed by the multiple linear regression analysis was statistically significant (p=0.000) for all the covariates (Table 3). The characteristics influenced system usage positively, with the exception of Mode of EHRs use characteristic. RDE mode of EHRs use had the highest negative impact on the use of the system. 

Table 3. Multiple linear regression model for staff system use and facility characteristics

Facility Characteristics Unstandardized Coefficients Standardized Coefficients t P-value

 B Std. Error Beta 

(Constant) 0.354 0.084 4.213 0.000

KEPH Level Level 2

Level 3

Level 4 

0.445

0.019 

0.194 

23.929 

0.000

Ownership -Faith-Based Organisation

-Ministry of Heath

-Non-Governmental Organization 

0.401 

0.035 

0.092 

11.308 

0.000

Services CT

CT&HTS 0.392 0.015 0.206 25.351 0.000

Mode of EHRs use Hybrid 

POC

RDE 

-0.124 

0.014 

-0.074 

-9.176 

0.000

Dependent Variable: Number of active system users; Independent Variables: KEPH level, ownership, mode EHRs of use, and services. p-value: when p<=0.05, there is statistically significant difference. B (coefficient) explains a change in dependent variable that can be attributed to a change of one unit in the independent variable.”

(Page 14/15, lines 297-307)

Discussion:

“Indeed, our study revealed that point of care (POC) and hybrid modes of data capture were associated with increased system usage compared to retrospective data entry. Thus, EHRs implementors should aim at point of care mode of operation right from initiation.” (Page 17, lines 358-361)

4. Overall, I believe that the paper does have interesting information to convey, and has potential for an analysis of this data that can make a contribution to the study of EHR in LMICs and in particular to studies of systems use. I do however suggest a further revision and edit to catch a number of small language issues and to strengthen the discussion of the data and its ability to reflect use. 

Response: Thank you for this feedback. Based on the above comment, we have completely reworked the manuscript in addressing language issues as shown in the track changes document. 

5. I would also tone down claims such as on page 18)to ‘conclusively give a true state of EHRs use’. That is a claim that no researcher across the world can make!! Equally I don’t think you can claim ‘highly reliable data’ (same page). I suggest fewer of such claims and more time addressing the subtlies of the data, set in the context. This will impress a reader far more.

Response: We have taken note of the comment and agree that the claims can be misleading. We have however revised that part of the manuscript as follows:

“Secondly, the study period (2012 – 2019) was long enough to reveal the state of the EHRs use in the health care facilities. Also, the study results are reliable due to the use of census method in the collection of the primary data.” (Page 17 lines 368-370) 

Reviewer #2:

Thank you for addressing the comments raised in a previous round of review and adjusting your manuscript. I believe the adjustments have strengthened the manuscript.

Response: We really appreciate the positive feedback on the revisions of our manuscript. Your comments have surely strengthened our manuscript.

1. Yet, I have one remark left about the adjustments made regarding the indicator ‘staff system use’. It was good to read you now address the issue of sharing individual-passwords. However, you very quickly draw the conclusion that training on appropriate use of account credentials is needed for staff. Thereby you assume that a lack of knowledge is the underlying cause, yet how do you know such a lack of knowledge exists? Can you refer to studies that indicate such a lack of knowledge? Other studies often point towards problems with the user-friendliness of EHRs, instead of a lack of knowledge among staff. Therefore, I believe you should look again if you can substantiate your conclusion/recommendation.

Response: We appreciate the reviewer’s pointing the need to support our claims. Our claim was informed by findings from focus group discussion with system users in the setting, which was part of this study. The publication on the findings in under review in another journal. However, we have revised that section of the manuscript as follows with relevant literature referenced:

“A study investigating users’ behaviour in password utilization revealed users share passwords for convenience as well as a show of trust [41]. The finding from our study warrants deeper assessment on user credentialing processes and account usage patterns (such as sharing of credentials). It also highlights the need to re-emphasize good password practices to the system users and active monitoring of user accounts by the system administrators. We also recommend further research to establish user-computer ratio in the healthcare facilities. (Page 15/16, lines 330-336)

We truly appreciate the feedback from the reviewers which have strengthened our manuscript. We hope that you will find our responses satisfactory. Thank you once again for considering our manuscript in PLOS One Journal.

Sincerely,

Philomena Ngugi

waruharip@gmail.com

Corresponding author

[1] W. H. DeLone and E. R. Mclean, “The DeLone and McLean Model of Information Systems Success: A Ten-Year Update,” J. Manag. Inf. Syst. / Spring, vol. 19, no. 4, pp. 9–30, 2003.

---

## [Editor Report · Decision Letter 2]

17 Aug 2021

A multivariate statistical evaluation of actual use of electronic health records systems implementations in Kenya

PONE-D-21-01911R2

Dear Dr. Ngugi,

We’re pleased to inform you that your manuscript has been judged scientifically suitable for publication and will be formally accepted for publication once it meets all outstanding technical requirements.

Kind regards,

Chaisiri Angkurawaranon

Academic Editor

PLOS ONE
---

## [Editor Report · Acceptance letter]

27 Aug 2021

PONE-D-21-01911R2 

A multivariate statistical evaluation of actual use of electronic health record systems implementations in Kenya 

Dear Dr. Ngugi:

I'm pleased to inform you that your manuscript has been deemed suitable for publication in PLOS ONE. Congratulations! Your manuscript is now with our production department. 

Kind regards, 

on behalf of

Dr. Chaisiri Angkurawaranon 

Academic Editor

PLOS ONE